# PUSH AND PULL: COMPETING FEATURE-PROTOTYPE INTERACTIONS IMPROVE SEMI-SUPERVISED SEMANTIC SEGMENTATION

## ABSTRACT

This paper challenges semi-supervised segmentation with a rethink on the feature-prototype interaction in the classification head. Specifically, we view each weight vector in the classification head as the prototype of a semantic category. The basic practice in the softmax classifier is to pull a feature towards its positive prototype (*i.e.*, the prototype of its class), as well as to push it away from its negative prototypes. In this paper, we focus on the interaction between the feature and its negative prototypes, which is always "pushing" to make them dissimilar. While the pushing-away interaction is necessary, this paper reveals a new mechanism that the contrary interaction of pulling close negative prototypes is also beneficial. We have two insights for this counter-intuitive interaction: 1) some pseudo negative prototypes might actually be positive so that the pulling interaction can help resisting the pseudo-label noises, and 2) some true negative prototypes might contain contextual information that is beneficial. Therefore, we integrate these two competing interactions into a Push-and-Pull Learning (PPL) method. On the one hand, PPL introduces the novel pulling-close interaction between features and negative prototypes with a feature-to-prototype attention. On the other hand, PPL reinforces the original pushing-away interaction with a multi-prototype contrastive learning. While PPL is very simple, experiments show that it substantially improves semi-supervised segmentation and sets a new state of the art.

## 1 INTRODUCTION

This paper considers the semi-supervised semantic segmentation task. We focus on an essential component in the segmentation model, *i.e.* the classification head, which consists of a set of learnable weight vectors. These weight vectors are usually viewed as a set of prototypes representing the corresponding semantic categories. The essential training process is to pull each deep feature towards its positive prototype (*i.e.*, the prototype of its class), as well as to push it away from its negative prototypes. The "pushing-away" interaction between a feature and its negative prototypes makes them dissimilar to each other and is critical for discriminating different classes. In all the following parts of this paper, **our discussion focuses on the interaction between features and their negative prototypes** and neglects the positive prototypes (unless explicitly pointed out).

While this "pushing away" interaction is necessary, this paper reveals a new mechanism that the contrary interaction of pulling close features and their negative prototypes is also beneficial. This "pulling close" interaction may seem counter-intuitive at the first glance but is actually reasonable from two insights as below:

1) It brings *a task-specific benefit* for semi-supervised segmentation by resisting the pseudo-label noises. Specifically, the popular pseudo-label-based pipeline is inevitably confronted with the pseudo-label noises: some pseudo negative prototypes might be actually positive. Under this condition, the "pulling close" interaction gives the feature a chance to approach its actual-positive prototype. Experiments confirm that the pulling interaction effectively reduces the pseudo label noises (Section 4.4) and this task-specific benefit is the primary reason for our improvement (Section 4.4).

2) It brings *a general benefit* for both the semi-supervised and fully-supervised segmentation because some negative prototypes contain contextual information. Specifically, since our prototypes (*i.e.*,

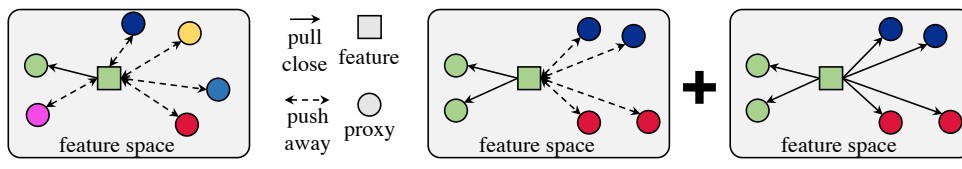

Figure 1: Comparison between the standard feature-prototype interaction and our push-and-pull learning. (a) In the standard classification head, the interaction with the negative prototypes is always "pushing-away". (b) This paper reveals that in addition to the pushing-away interaction, a competing pulling-close interaction between features and negative prototypes is also beneficial. Therefore, the proposed push-and-pull learning (PPL) combines the pushing-away and pulling-close interactions for negative prototypes. Moreover, we use multiple prototypes to represent each individual class to better reflect the intra-class diversity, which is also beneficial. Circles and squares represent prototypes and features, respectively. Different classes are in different colors.

the weight vectors) are learned from the whole training set through back-propagation, they naturally provide clues for deriving the contextual information. Some prior works (Yuan et al., 2020; Jin et al., 2021; Zhou et al., 2022) already show that exploring the contextual information from mean features improves the segmentation accuracy. In contrast, we find that directly using the weight vectors for contextual information can bring the similar benefit and is relatively simple. Empirically, we show this general benefit is a secondary reason for our improvement (as detailed in Section 4.4).

These two insights motivate us to propose Push-and-Pull Learning (PPL) for semi-supervised segmentation, as shown in Fig. 1. **O**n the one hand, PPL reinforces the basic "pushing negative prototypes" interaction with a multi-prototype contrastive learning, *i.e.*, using multiple (instead of one) prototypes to better represent each category. **O**n the other hand, PPL enforces the "pulling negative prototypes" interaction through a feature-to-prototype attention. Specifically, each feature absorbs information from all the prototypes using a standard cross-attention layer. Although these two interactions share a same effect of pulling close positive prototypes, we name the resulting method as Push-and-Pull Learning to highlight their competing effect on the negative prototypes.

Another important advantage of the proposed PPL is that it can be cascaded to enlarge its benefit. It is because, after the pulling-close interaction, the features are refined with the improved resistance against pseudo-label noises. Therefore, we re-evaluate their pseudo labels to increase label accuracy. Based on the refined deep features, we may append another round of PPL for further improvement. Extensive experiments on popular benchmarks validate that cascading multiple PPL accumulates multiple rounds of improvement (Appendix B.2) and sets a new state of the art (*e.g.*, 77.03% and 74.22% mIoU on PASCAL VOC 2012 using 1/8 and 1/16 labels). Moreover, ablation study shows that within the two benefits of PPL, the task-specific benefit is the major reason for the superiority of PPL: while the general benefit improves the fully-supervised segmentation (the upper-bound of semi-supervised segmentation) by a small margin, the benefit of resisting pseudo-label noises largely reduces the gap between the few-label and full-label regimes.

To sum up, this paper makes the following contributions:

• We investigate the interactions between features and negative prototypes for semi-supervised segmentation and reveal a novel mechanism, *i.e.*, while the standard pushing-away interaction is necessary, the contrary interaction of pulling close negative prototypes is also beneficial.

• We correspondingly propose a pull-and-push learning (PPL) method. PPL combines two competing interactions and brings two benefits, *i.e.*, resisting the pseudo-label noises and leveraging the contextual information. Moreover, cascading multiple PPL stages can accumulate its benefits and thus enlarge the improvement.

• We empirically show that PPL substantially improves semi-supervised segmentation by reducing the accuracy gap between full supervision and semi supervision. The achieved results set new state of the art on two popular benchmarks.

## 2 RELATED WORK

### 2.1 SEMI-SUPERVISED SEMANTIC SEGMENTATION

Recently, consistency regularization and pseudo-labeling have been extensively explored in semi-supervised segmentation (French et al., 2019; Ouali et al., 2020; Ke et al., 2020; Chen et al., 2021; He et al., 2021; Hu et al., 2021; Liu et al., 2022), especially after the success of Mean Teacher (Tarvainen & Valpola, 2017) and FixMatch (Sohn et al., 2020) in semi-supervised classification. For example, French et al. (2019) design consistency supervision based on CutMix augmentation to exploit unlabeled data while Ouali et al. (2020) and Ke et al. (2020) consider feature and network perturbations, respectively, when designing consistency regularization for unlabeled data. He et al. (2021) propose to improve model training by producing unbiased pseudo labels which match the true class distribution in labeled data. Hu et al. (2021) design a confidence bank to balance the training of well and badly performed categories. Liu et al. (2022) extend the mean-teacher model for semi-supervised segmentation by including a new auxiliary teacher and developing a stricter confidence-weighted cross-entropy loss. In this work, we also use consistency regularization and pseudo-labeling to exploit unlabeled data and propose pushing and pulling feature-prototype interactions to improve the network training for semi-supervised segmentation.

### 2.2 CONTRASTIVE LEARNING

Contrastive learning aims to pull positive pairs close while pushing negative pairs far away from each other in the feature space. Recently, contrastive learning has shown its great potential to exploit unlabeled data in self-supervised learning (Hjelm et al., 2018; Wu et al., 2018; He et al., 2020; Chen et al., 2020). This inspires the exploration of contrastive learning in semi-supervised segmentation (Zhou et al., 2021; Alonso et al., 2021; Zhong et al., 2021; Lai et al., 2021; Wang et al., 2022). For example, Lai et al. (2021) propose to establish consistency between the pixels at the same location but with different contexts and design a directional contrastive loss with the help of pseudo labels. Alonso et al. (2021) maintain a memory bank to save high-quality feature vectors from labeled data and perform contrastive learning with this memory bank to improve the network training. Wang et al. (2022) propose to effectively use unreliable pseudo labels with contrastive learning. These state-of-the-art methods usually consider pixel-to-pixel contrastive learning. In contrast, the contrastive learning part in our method compares features against prototypes. We represent each class with multiple prototypes so as to reflect intra-class diversity. Therefore, it better accommodates the mixture of labeled and unlabeled data and is one important reason for the superiority of our method.

### 2.3 CONTEXT LEARNING

Context learning is a popular way to improve semantic segmentation by exploring contextual information from pixels or regions. Yuan et al. (2020) propose OCRNet to augment features with the object-contextual representations from local contexts. Jin et al. (2021) leverage object-contextual representations from the whole datasets to improve features. Zhou et al. (2022) apply context learning to weakly supervised semantic segmentation. In this work, our PPL provides an alternative approach for context learning, *i.e.*, directly using the weight vectors. However, we note that context learning is not our novelty and is merely the secondary reason for our improvement.

## 3 METHODOLOGY

The overview of Push-and-Pull Learning (PPL) is illustrated in Fig. 2. PPL cascades multiple stages for successive improvement (Fig. 2(a)). Each PPL stage consists of 1) a standard classifier for segmentation (with the cross-entropy loss), 2) a contrast-based pushing interaction (Fig. 2(b)) and 3) an attention-based pulling interaction (Fig. 2(c)). Both the contrast-based pushing interaction and the attention-based pulling interaction require storing the historical prototypes from the classifier into a classifier memory bank CM, so that each class has multiple prototypes. After the attention-based pulling interaction, the features are refined to some extent and are fed into the classifier of the sub-sequential PPL stage as its input. We note that although our method cascades multiple PPL stages to refine the pseudo labels, it only introduces slight computational overhead (as detailed in Appendix C).

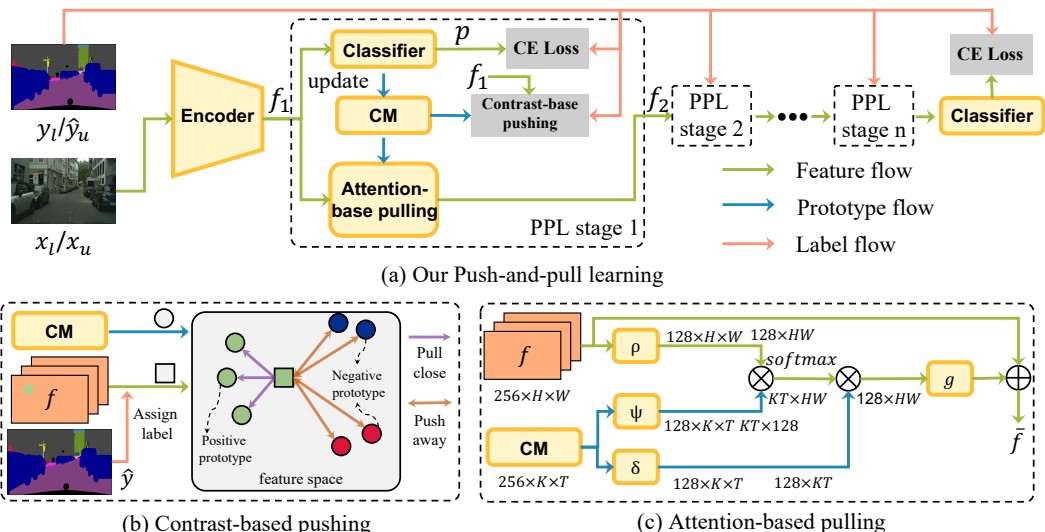

Figure 2: Overview of the push-and-pull learning. We perform the pushing interaction by multi-prototype contrastive learning, as shown in (b), and the pulling interaction via a feature-to-prototype attention, as shown in (c). The classifier memory bank CM is designed as a queue to store class prototypes and is updated by the corresponding classifier weights. Multiple pulling and pushing interactions are leveraged sequentially to enlarge their benefit. "$\otimes$" and "$\oplus$" in (c) denote matrix multiplication and element-wise sum, respectively. $\rho$, $\psi$, $\delta$ and $g$ are four convolutional modules.

Section 3.1 and Section 3.2 elaborate on the contrast-based pushing and the attention-based pulling, respectively. Section 3.3 illustrates how to integrate the proposed PPL into a popular teacher-student framework for semi-supervised segmentation.

## 3.1 CONTRAST-BASED PUSHING INTERACTION

The contrast-based pushing interaction uses multiple prototypes to represent each single semantic class and complements the standard classifier (single prototype for each class). The motivation: using a single prototype lacks consideration for the intra-class diversity and is prone to over-fitting the noisy pseudo-labeled samples in the current minibatch. Therefore, we collect additional prototypes from historical classifiers (*i.e.*, the classifiers in previous iterations) and store them in a classifier memory bank CM, as illustrated in Fig. 2(a). Specifically, CM is updated in a first in last out manner and maintains a constant size of $K \times T \times C$ ($K$ is the total number of classes, $T$ is the number of prototypes per class and $C$ is the dimension of each prototype). CM improves the robustness and effectiveness of pushing-away interaction through a multi-prototype contrastive loss function:

$$
\boldsymbol{L}_{cont}(f, \text{CM}, \hat{y}, \mathcal{M}) = \frac{1}{||\mathcal{M}||_1} \sum_{h=1}^{H} \sum_{w=1}^{W} \mathcal{M}^{h,w} \log(1 + \sum_{t=1}^{T} \exp(\gamma(-\phi(f^{h,w}, \text{CM}^{\hat{y}^{h,w},t}))))
$$
$$
\sum_{k=1, k \neq \hat{y}^{h,w}}^{K} \sum_{t=1}^{T} \exp(\gamma(\phi(f^{h,w}, \text{CM}^{k,t}) + m)),
$$

(1)

where $H$ and $W$ denote the height and width of the feature map, respectively. $\mathcal{M} \in \{0,1\}^{H \times W}$ is a mask for selecting high-confident pseudo-labelled pixels for training. $\phi(a,b) = \frac{a \cdot b}{||a||_2 \times ||b||_2}$ denotes the cosine similarity between a feature vector and a prototype. $f^{h,w}$ and $\hat{y}^{h,w}$ denote the feature vector and the corresponding pseudo label at the location $(h, w)$, respectively. $\text{CM}^{k,t}$ denotes the prototype of the $k$-th class in the $t$-th previous iteration. $\gamma$ and $m$ are the scale factor and the margin, respectively. In Eq. (1), the first component aims to pull one pixel feature $f^{h,w}$ close to its positive prototypes $\text{CM}^{\hat{y}^{h,w},t}$ (belonging to the same class ($\hat{y}^{h,w}$) as the feature) while the second component is designed to push the feature far away from its negative prototypes.

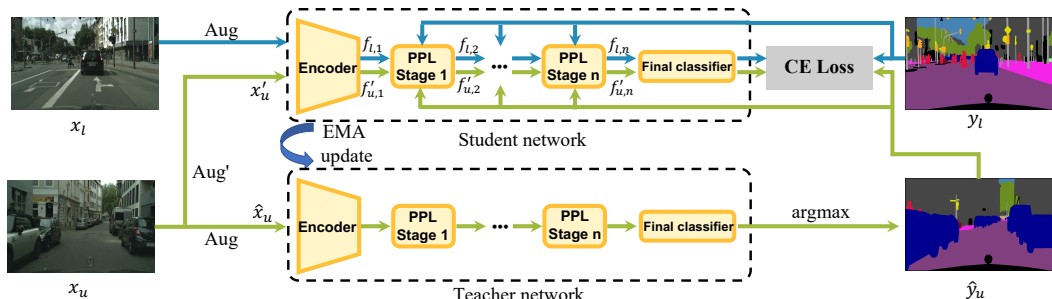

Figure 3: Integrating the proposed PPL into a popular teacher-student framework for semi-supervised segmentation. The student and teacher networks share the same architecture and both contain multiple ($n$) PPL stages. The detailed structure of each PPL stage is illustrated in Fig. 2(a). Please note that while each PPL has its own classifier, there is one more classifier ("Final classifier") after the last PPL stage. This final classifier uses the refined features of the last PPL as its input features. The student network uses the ground truth on the labeled data and the pseudo labels on the unlabeled data for supervision. The teacher network applies weak augmentation (Aug) to the unlabeled data, while the student network mixes both weak and strong augmentation (Aug').

## 3.2 ATTENTION-BASED PULLING INTERACTION

Given the prototypes in CM and the features, the proposed PPL incorporates a pulling-close interaction to compete with the contrast-based pushing interaction. This pulling-close interaction is implemented with a feature-prototype attention as shown in Fig. 2(c). Specifically, we first compute the affinity between each feature $f^{h,w}$ and each prototype $\text{CM}^{k,t}$, which is defined as,

$$w_{hw,kt} = \frac{\exp(\rho(f^{h,w})^{\top}\psi(\text{CM}^{k,t}))}{\sum_{k'=1}^{K}\sum_{t'=1}^{T}\exp(\rho(f^{h,w})^{\top}\psi(\text{CM}^{k',t'}))}, \tag{2}$$

where $w_{hw,tk}$ is the feature-prototype affinity score, $\rho$ and $\psi$ are two convolutional blocks. Each block consists of two "$1 \times 1$ convolutional layers + batch normalization + ReLU layer".

According to the affinity scores, each feature absorbs some information from all the prototypes into itself for the pulling-close interaction, which is formulated as:

$$\bar{f}^{h,w} = (g(\sum_{k=1}^{K}\sum_{t=1}^{T} w_{hw,kt}\delta(\text{CM}^{k,t})) + f^{h,w})/2, \tag{3}$$

where $g$ and $\delta$ are two convolution blocks consisting of a $1 \times 1$ convolutional layer, a sub-sequential batch normalization and a ReLU layer. $\bar{f}^{h,w}$ denotes the refined feature vector at the location $(h, w)$ after absorbing information from the prototypes.

**Discussion.** The pulling interaction between features and prototypes can be also viewed as an interaction among all the labeled and unlabeled data across the whole training set. This is because the class prototypes are learned from both the labeled and pseudo-labeled samples, simultaneously. Therefore, when a feature absorbs information from these prototypes (Eq. (3)), it constructs a pulling-close interaction towards all its inherently-similar samples in the training set. Under the case where an unlabeled sample is assigned with an incorrect pseudo label, this interaction is particularly beneficial because it allows the feature to approach its ground-truth class prototype, as well as its true-positive samples. Consequently, the pulling interaction facilitates resistance against noisy pseudo labels and helps refine the features. Using the oracle view, we find that using the refined features to assign the pseudo labels improves the pseudo label quality, as detailed in the supplementary.

## 3.3 INTEGRATING PPL INTO SEMI-SUPERVISED TEACHER-STUDENT FRAMEWORK

We show how to integrate the proposed PPL into a popular teacher-student framework for semi-supervised segmentation, as illustrated in Fig. 3. We first present the teacher-student framework as follows. The student and teacher models share the same network architecture and are parameterized

with $\theta$ and $\theta'$, respectively. The teacher model $\theta'$ is updated as the exponential moving average (EMA (Tarvainen & Valpola, 2017)) of the student model $\theta$, which is formulated as:

$$\theta'_t = \alpha \cdot \theta'_{t-1} + (1 - \alpha) \cdot \theta_t, \tag{4}$$

where $\alpha$ is a coefficient controlling the update rate.

In the teacher-student framework, the labeled samples and unlabeled samples are jointly used for training the student. Specifically, the labeled samples $x_l$ (after data augmentation) are directly fed into the student network for training. In contrast, the unlabeled samples $x_u$ are fed into both student and teacher networks in parallel, because we require the teacher prediction for generating the pseudo labels to supervise the student. We note that the student and teacher networks have different data augmentations for the unlabeled data: the teacher network prefers relatively weak data augmentation and transfers $x_u$ to $\hat{x}_u$, while the student mixes both weak and strong augmentation and transfers $x_u$ to $x'_u$. This data augmentation strategy is consistent with FixMatch (Sohn et al., 2020).

We integrate the proposed PPL into this training framework by inserting our cascaded PPL stages between the encoder (feature extractor) and the classifier, as shown in Fig. 3, for both the student and teacher networks. We note that each PPL stage has its own classifier (as shown in Fig. 2 (a)), and there is one more classifier ("Final classifier" in Fig. 3) after the last PPL stage. For the student network, all these classifiers need to be learned on the combination of labeled and unlabeled data. Specifically, for the labeled data, we employ the cross-entropy loss for supervising the student predictions, which is formulated as:

$$\boldsymbol{L}_l = -\frac{1}{H \times W} \sum_{i=1}^{n+1} \sum_{h=1}^{H} \sum_{w=1}^{W} y_l^{h,w} \log(p_{l,i}^{h,w}), \tag{5}$$

where $y_l^{h,w}$ denotes the ground-truth of one labeled image at the location $(h, w)$ and $p_{l,i}^{h,w}$ denote the corresponding prediction. $i = 1, ..., n$ indicates the prediction is predicted in the $i$-th PPL stage while $i = n + 1$ indicates the prediction is from the final classifier. $n$ is the number of PPL stages.

As for the unlabeled data, given the pseudo labels from the teacher, we apply both cross-entropy loss and the proposed contrastive loss (Eq. (1)) for supervising the student predictions, which is formulated as:

$$\boldsymbol{L}_u = -\frac{1}{||\mathcal{M}||_1} \sum_{i=1}^{n+1} \sum_{h=1}^{H} \sum_{w=1}^{W} \mathcal{M}^{h,w} \hat{y}_u^{h,w} (\log(p_{u,i}^{h,w})) + \sum_{i=1}^{n} \boldsymbol{L}_{cont}(f_{u,i}, \mathrm{CM}_i, \hat{y}_u, \mathcal{M}), \tag{6}$$

where $f_{u,i}$ and $p_{u,i}$ denote the feature before the classifier and the corresponding prediction of one weakly or strongly augmented unlabeled image from the $i$-th PPL stage. $\hat{y}_u$ is a one-hot pseudo label map generated according to the prediction $\hat{p}_u$ from the teacher network. $\mathcal{M} = \mathbb{1}(\max(\hat{p}_u) > \tau)$ aims to select reliable pseudo labels for training with a threshold $\tau = 0.8$.

The overall loss consists of losses for the labeled and unlabeled data, *i.e.*, $\boldsymbol{L}_l$ and $\boldsymbol{L}_u$, written as,

$$\boldsymbol{L} = \boldsymbol{L}_l + \lambda \boldsymbol{L}_u, \tag{7}$$

where $\lambda$ is a coefficient which controls the contributions of different losses. The teacher network is updated as the momentum of the student network and thus requires no supervision.

## 4 EXPERIMENTS

### 4.1 DATASETS AND EVALUATION

**Datasets:** We conduct our experiments on two standard segmentation datasets, *i.e.*, PASCAL VOC 2012 (Everingham et al., 2015) and Cityscapes (Cordts et al., 2016). PASCAL VOC 2012 contains 21 semantic classes to segment (including the background class). In PASCAL VOC 2012 (the augmented set (Hariharan et al., 2011)), $10, 582$, $1, 449$ and $1, 456$ images are used for training, validation and test, respectively. Cityscapes focuses on urban scene understanding and contains 19 classes to segment. In Cityscapes, $2, 975$, $500$ and $1, 525$ images are used for training, validation and test, respectively. In our semi-supervised segmentation experiments, we randomly sample 1/2,

Table 1: Comparison with state-of-the-art methods on PASCAl VOC 2012. Two backbones, *i.e.*, ResNet-50 and ResNet-101 and four data partition protocols, *i.e.*, 1/16 , 1/8 , 1/4 and 1/2 are considered. The results of U²PL(Wang et al., 2022) are from the authors' GitHub issues.

| Methods | ResNet-50 (%) | | | | ResNet-101 (%) | | | |
|---|---|---|---|---|---|---|---|---|
| | 1/16 | 1/8 | 1/4 | 1/2 | 1/16 | 1/8 | 1/4 | 1/2 |
| MT (Tarvainen & Valpola, 2017) | 66.77 | 70.78 | 73.22 | 75.41 | 70.59 | 73.20 | 76.62 | 77.61 |
| CCT (Ouali et al., 2020) | 65.22 | 70.87 | 73.43 | 74.75 | 67.94 | 73.00 | 76.17 | 77.56 |
| CutMix-Seg (French et al., 2019) | 68.90 | 70.70 | 72.46 | 74.49 | 72.56 | 72.69 | 74.25 | 75.89 |
| GCT (Ke et al., 2020) | 64.05 | 70.47 | 73.45 | 75.20 | 69.77 | 73.30 | 75.25 | 77.14 |
| CPS (Chen et al., 2021) | 71.98 | 73.67 | 74.90 | 76.15 | 74.48 | 76.44 | 77.68 | 78.64 |
| U²PL(Wang et al., 2022) | - | - | - | - | 74.43 | 77.60 | 78.70 | 79.94 |
| PS-MT (Liu et al., 2022) | 72.83 | 75.70 | 76.43 | 77.88 | 75.50 | 78.20 | 78.72 | 79.76 |
| PPL (Ours) | **74.22** | **77.03** | **77.33** | **78.03** | **77.10** | **78.57** | **79.50** | **80.20** |

1/4, 1/8 and 1/16 of the whole training set as labeled images and treat the rest as unlabeled images. We use the same split lists as (Chen et al., 2021) for both two datasets.

**Evaluation:** We leverage mean Intersection-over-Union (mIoU) metric to measure the segmentation performance. Following (Chen et al., 2021), we report results on 1,456 PASCAL VOC 2012 val set (or 500 Cityscapes val set) on one single scale testing.

## 4.2 IMPLEMENTATION DETAILS

In this work, we leverage DeepLabv3+ (backboned on ImageNet-pretrained ResNet-50 and ResNet-101 (He et al., 2016)) as our baseline network. The encoder in Fig. 2(a) denotes all components of DeepLabv3+ except the classifier, including the backbone, ASPP and decoder module.

During training, we adopt random scaling, cropping and flipping for both labeled and unlabeled data. For the unlabeled data, we conduct strong augmentation by additionally applying GrayScale, GaussianBlurring, and ColorJitter. Stochastic gradient descent (SGD) with momentum set to $0.9$ is leveraged to optimize the network, and the weight decay is set to $1e^{-4}$. Besides, we adopt the "ploy" learning rate policy where the initial learning rate is multiplied by $(1 - \frac{iter}{max\_iter})^p$ with $p = 0.9$. The learning rate for the pretrained backbone is additionally multiplied by $0.1$. The length of the classifier memory bank is set to 10 and the number $n$ of our PPL stages is set to 2. $\gamma$ and $m$ in Eq. (1) is set to 32 and 0.25, respectively. The updating rate $\alpha$ for the teacher network in Eq. (4) changes during training, written as $\alpha = 0.99 + 0.01(1 - \cos(\pi \cdot \frac{iter}{max\_iter} + 1)/2)$.

On PASCAL VOC 2012, we randomly crop images into $512 \times 512$ images and train our network for 80 epochs with the base learning rate set to $0.01$. In the first 10 epochs, we warm up our network by only using labeled data. We set the batch size to 8 and the loss coefficient $\lambda$ in Eq. (7) is set to $0.1$. The iteration number per epoch is measured by the whole training set size and the batch size. On Cityscapes, we randomly crop images into $800 \times 800$ images and train networks for 240 epochs with the base learning rate set to $0.1$. We use 20 epochs to warm up our network. We set the batch size to 8 and the loss coefficient $\lambda$ to $0.5$. Besides, OHEM loss is used for labeled images and CutMix augmentation is used for unlabeled images on Cityscapes.

## 4.3 COMPARISON WITH STATE-OF-THE-ART METHODS

Table 1 and Table 2 compare the proposed PPL with seven state-of-the-art methods on PASCAL VOC 2012 and Cityscapes, respectively. From these comparisons, we draw two observations as below:

First, PPL achieves accuracy on par with the state of the art. For example, on Pascal VOC 2012, PPL based on ResNet-50 surpasses the strongest competitor PS-MT (Liu et al., 2022) by 1.39%, 1.33%, 0.90% and 0.15% under 1/16, 1/8, 1/4 and 1/2 data partition protocols, respectively.

Second, focusing on the accuracy decrease from "1/2" to "1/16" data partition, we observe that PPL presents slower decreases, compared with the competing methods. For example, on Pascal VOC 2012, the competing methods suffer decreases of 4.17% (CPS) and 5.05% (PS-MT) when the labeled data is reduced from "1/2" to "1/16". In contrast, given the same condition, the accuracy decrease

Table 2: Comparison with state-of-the-art methods on Cityscapes. Two backbones, *i.e.*, ResNet-50 and ResNet-101 and four data partition protocols, *i.e.*, 1/16, 1/8, 1/4 and 1/2 are considered.

| Methods | ResNet-50 (%) | | | | ResNet-101 (%) | | | |
|---|---|---|---|---|---|---|---|---|
| | 1/16 | 1/8 | 1/4 | 1/2 | 1/16 | 1/8 | 1/4 | 1/2 |
| MT (Tarvainen & Valpola, 2017) | 66.14 | 72.03 | 74.47 | 77.43 | 68.08 | 73.71 | 76.53 | 78.59 |
| CCT (Ouali et al., 2020) | 66.35 | 72.46 | 75.68 | 76.78 | 69.64 | 74.48 | 76.35 | 78.29 |
| GCT (Ke et al., 2020) | 65.81 | 71.33 | 75.30 | 77.09 | 66.90 | 72.96 | 76.45 | 78.58 |
| CPS (Chen et al., 2021) | 74.47 | 76.61 | 77.83 | 78.77 | 74.72 | 77.62 | 79.21 | 80.21 |
| PS-MT (Liu et al., 2022) | - | 77.12 | 78.38 | 79.22 | - | - | - | - |
| PPL (Ours) | **77.02** | **78.54** | **79.72** | **80.29** | **77.54** | **78.84** | **80.21** | **80.79** |

Table 3: Effectiveness of the attention-based pulling interaction and the contrast-based pushing interaction. "Pulling" and "Pushing" denote the developed pulling and pushing interactions, respectively. "Semi" and "Fully" denotes networks are trained in semi-supervised and fully-supervised manners, respectively, while "Gap" denotes the accuracy gap between "Semi" and "Fully". Results with semi-supervised learning are evaluated on PASCAL VOC 2012 under $1/8$ partition protocol.

| Method | Pulling | Pushing | ResNet-50 (%) | | | ResNet-101 (%) | | |
|---|---|---|---|---|---|---|---|---|
| | | | Semi | Fully | Gap | Semi | Fully | Gap |
| Baseline | | | 72.15 | 75.71 | 3.56 | 75.35 | 78.68 | 3.33 |
| PPL | ✓ | | 74.40 | 77.00 | 2.60 | 77.35 | 79.63 | 2.28 |
| | | ✓ | 74.15 | 76.84 | 2.69 | 76.33 | 79.21 | 2.88 |
| | ✓ | ✓ | **77.03** | **78.05** | **1.02** | **78.57** | **80.64** | **2.07** |

of the proposed PPL is relatively small, *i.e.*, $-3.81\%$. Therefore, PPL reduces the gap between abundant labeled data and very few labeled data. We recall that PPL has two benefits for semi-supervised segmentation, *i.e.*, a general one (*i.e.*, contextual information for general segmentation) and a task-specific one (resistance against pseudo-label noises). This observation shows that the task-specific benefit is the main reason for the superiority of PPL.

## 4.4 Effectiveness of the pushing and pulling interactions

We investigate the pushing and pulling interactions through ablation study under the semi-supervised ($1/8$ labeled data) and the fully-supervised scenario. The results are summarized in Table 3, from which we draw three observations: First, both the pulling and pushing interactions bring independent improvement to the baseline. For example, based on ResNet-50, the pushing and pulling interaction improve the baseline by $2.00\%$ and $2.25\%$, respectively.

Second, combing the pushing and pulling interactions achieves further improvement. For example, based on ResNet-50, PPL achieves a $4.89\%$ mIoU overall improvement. It indicates that these two interactions are complementary to each other.

Third, comparing "Semi" against "Fully", we notice PPL reduces the "Gap" between semi-supervision and full supervision. For example, the baseline undergoes a 3.56% gap, while PPL reduces this gap to 1.02%. It is consistent with our second observation in Section 4.3, further confirming that the superiority of PPL mainly comes from its task-specific benefit for semi-supervision.

## 4.5 PPL improves the pseudo label quality

PPL enables to improve the pseudo label quality. We evaluate the generated pseudo labels during training and present the quantitative and qualitative results in Fig. 4 and Fig. 5, respectively. In particular, three methods are considered, as follows: 1) "baseline" (the popular teacher-student framework), 2) "PPL one stage" (adding one PPL stage into the baseline) and 3) "PPL two stage" (Ours). In Fig. 4, the used pseudo labels are collected and evaluated per epoch, and mIoU and averaged recall are used to measure the quality of the pseudo labels.

We draw two observations from Fig. 4. First, we observe that PPL is able to improve both mIoU and recall of pseudo-labeled pixels. It indicates that our PPL is able to incorporate more pixels with

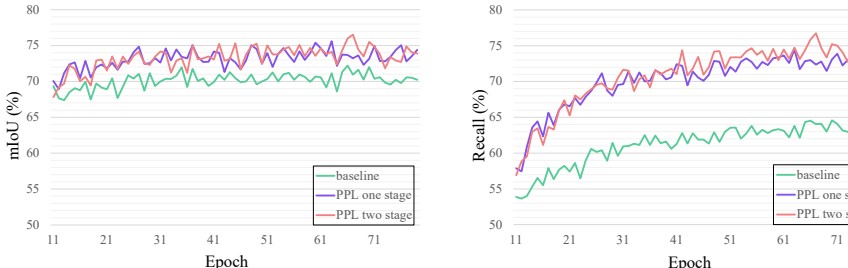

Figure 4: Quantitative comparison of generated pseudo labels from 1) the popular teacher-student "baseline", 2) "PPL one stage" (using one PPL stage) and 3) "PPL two stage" (Ours).

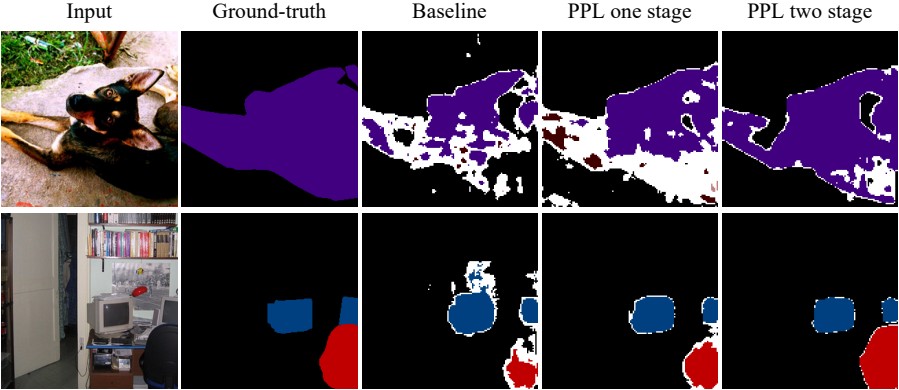

Figure 5: Qualitative comparison of the pseudo label quality. Some pixels are in white color because the predicted pseudo labels are filtered out due to low prediction confidence.

more accurate pseudo labels into training. Second, in Fig. 4, we also notice adding one more PPL stage improves the recall sightly (while maintaining the mIoU). In particular, we average the mIoU and recall from all epochs for "PPL one stage" and "PPL two stage". "PPL one stage" achieves 73.16% in the averaged mIoU and 70.26% in the averaged recall, while "PPL two stage" achieves 73.27% and 70.93% in the averaged mIoU and recall, respectively.

Combining the above two observations, we infer that PPL refined the features and thus improves the pseudo label quality and this refining effect can be enlarged by cascading multiple PPL stages.

Fig. 5 visualizes the pseudo labels generated from the baseline, one-stage PPL and two-stage PPL, respectively. Different colors denote different classes while white indicates the pseudo labels are filtered out by the threshold $\tau$ and are not considered during training. The visualization is consistent with the quantitative results.

## 5 CONCLUSION

This paper proposes a novel push-and-pull learning (PPL) for semi-supervised semantic segmentation. In contrast to the common practice of pushing features away from their negative prototypes, PPL considers pulling features close to some negative prototypes is sometimes beneficial, as well. Therefore, PPL integrates these two competing feature-prototype interactions for cooperation. On the one hand, PPL reinforces the fundamental pushing interaction with a multi-prototype contrastive learning. On the other hand, PPL uses a feature-to-prototype attention to make the feature absorb information from the negative prototypes, yielding the pulling effect. These two interactions both improve the semi-supervised baseline, and bring further improvement when PPL combines them together. Extensive experiments show that PPL effectively reduces the accuracy gap between semi-supervision and full-supervision, and the achieved results are on par with the state of the art.

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

## A   PPL IMPROVES THE ROBUSTNESS AGAINST NOISY PSEUDO LABELS

Table 4: Comparison between our PPL method and baseline with noisy pseudo labels. "mIoU Gap" denotes the mIoU gap between situations with ground-truth and with noisy pseudo labels.

| Noise ratio | PPL (%) | | baseline (%) | |
| --- | --- | --- | --- | --- |
| | mIoU | mIoU Gap | mIoU | mIoU Gap |
| 0% | 78.37 | - | 74.69 | - |
| 10% | 77.74 | -0.63 | 73.90 | -0.79 |
| 20% | 77.67 | -0.70 | 72.94 | -1.75 |
| 40% | 76.35 | -2.02 | 71.17 | -3.75 |

To explicitly show that our PPL can resist pseudo-label noises, we manually add noises into the training process. Specifically, during training, we simulate experiments under the 1/8 partition protocol, *i.e.*, sampling 1/8 data of the whole training set as the labeled samples and using the remaining 7/8 data as the unlabeled samples. For the unlabeled data, we use oracle view (ground truth) to set up the conditions with clean labels, *i.e.* there is 0% pseudo label noise. We set up three noisy conditions, *i.e.*, there are 10% / 20%/ 40% pseudo label noises, by replacing ground-truth with inaccurate pseudo labels at the corresponding ratios. To be specific, we randomly select the index corresponding to the 2nd or the 3rd largest softmax prediction of the teacher network to replace the ground

truth. Since the 2nd and the 3rd largest indexes correspond to the "easy-to-confuse" classes, using them as the incorrect pseudo labels are reasonable.

The results are summarized in Table 4, from which we clearly observe that when the pseudo labels become noisy, PPL has less accuracy drop than the baseline. For example, when the ratio of noisy pseudo labels is 10%, 20% and 40%, the accuracy drop of the baseline is -0.79%, -1.75% and -3.75%, respectively. In contrast, the accuracy drop of the proposed PPL is smaller (-0.63%, -0.70% and -2.02%, respectively). Specifically, the superiority of PPL is more significant when the noise ratio is larger. *This observation confirms that PPL improves the robustness against noisy pseudo labels.*

In Table 4, "0% noise" achieves different accuracy from "Fully" in Table 3 (main text) because they have different experiments setup. In "Fully" experiments, all data instead of partial data are used for both supervised training and consistency based unsupervised training. Besides, in "Fully" experiments, pseudo labels rather than ground-truth are used for the consistency-based unsupervised training.

# B IMPACT OF HYPER-PARAMETERS

## B.1 IMPACT OF THE LENGTH OF CLASSIFIER MEMORY BANKS ($T$)

We recall that PPL collects multiple ($T$) historical prototypes into the memory bank, therefore better reflecting the intra-class diversity. Table 5(a) investigates the impact of $T$ with a ResNet-50-based network under the 1/8 partition protocol on PASCAL VOC 2012, from which we observe that the accuracy first gradually increases (when $T$ increases from 1 to 10) and then decreases. It is reasonable because when $T$ is too large, some prototypes are severely out-of-date and actually become distractors. Therefore, we recommend using $T = 10$ as an optimum.

## B.2 IMPACT OF THE NUMBER OF PPL STAGES ($n$)

In Table 5(b), we conduct experiments to analyse the impact of the number of PPL stages ($n$). As shown in Table 5(b), PPL achieves the best accuracy when $n$ is set to 2. We infer it is because using fewer stages cannot take the full advantage of PPL while using too many stages may distort the output feature which involves too much class-wise prototype information. Therefore, we set $n = 2$ in all our experiments.

Table 5: Analysis of the impact of classifier memory bank length $T$ and PPL stage number $n$.

| $T$ | mIoU (%) |
|-----|----------|
| 1   | 75.86    |
| 5   | 76.95    |
| 10  | 77.03    |
| 20  | 76.20    |

(a)

| $n$ | mIoU (%) |
|-----|----------|
| 0   | 72.15    |
| 1   | 76.16    |
| 2   | 77.03    |
| 3   | 76.85    |
| 4   | 76.18    |

(b)

## B.3 IMPACT OF THE SCALE FACTOR $\gamma$ AND THE MARGIN $m$

We recall that the scale factor $\gamma$ and the margin $m$ are adopted in the developed contrastive loss (in Eq. (1)) for better similarity separation. Table 6 investigates the impact of $\gamma$ and $m$ with a ResNet-50-based network under the 1/8 partition protocol on PASCAL VOC 2012. As shown in Table 6, better accuracy is achieved when $\gamma$ and $m$ are set to 32 and 0.25, respectively. Thus, we set $\gamma$ to 32 and $m$ to 0.25 in our experiments.

# C COMPUTATIONAL COST OF PPL.

We recall that PPL cascades multiple (2) stages which bring additional computational cost during inference. Thus, we report the number of parameters and GFLOPs of PPL stages in Table 7, and find that the computational cost of PPL stages are affordable.

Table 6: Analysis of the impact of the scale factor $\gamma$ and the margin $m$

| $\gamma$ | mIoU (%) |
|---|---|
| 16 | 76.14 |
| **32** | **77.03** |
| 64 | 75.19 |

(a)

| $m$ | mIoU (%) |
|---|---|
| 0.15 | 75.80 |
| **0.25** | **77.03** |
| 0.35 | 76.63 |

(b)

Table 7: Computational cost of PPL during inference.

| Method | #params (M) | FLOPs (G) |
|---|---|---|
| wo/ PPL | 40.42 | 76.24 |
| One PPL stage | 40.58 | 77.60 |
| Two PPL stage | 40.75 | 78.96 |

