# OpenReview forum: "Push and Pull:  Competing Feature-Prototype Interactions  Improve Semi-supervised Semantic Segmentation"
_ICLR.cc/2023/Conference — Submitted to ICLR 2023_

### Official Review · Reviewer_ewoj · 2022-10-24

**Confidence:** 4
**Correctness:** 4
**Technical Novelty And Significance:** 3
**Empirical Novelty And Significance:** 3
**Recommendation:** 5

**Clarity, Quality, Novelty And Reproducibility:**

Clarity: This paper adopts the prototype to represent the class weights, and I suggest adding a reference to the prototype paper [1] will be better.
[1] Zhou et.al. Rethinking Semantic Segmentation: A Prototype View. CVPR 2022.

Quality: The writing/presentation quality of this paper is clear and good.

Novelty: This paper proposes to explore the interaction between the pixel features and negative prototypes for better robustness and context for semi-supervised semantic segmentation. The motivation is somewhat attractive in this field. However, the proposed techniques have some relation to previous methods for semantic segmentation, limiting the novelty of this paper.

Reproducibility: There might be some difficulties in reproducing the method and experiments of this paper due to the lack of detailed settings of the semi-supervised part.


**Strength And Weaknesses:**

* Strength
1. This paper explores the impact of the interaction between the pixel features and negative prototypes in semantic segmentation and presents a Push-and-Pull Learning mechanism.
2. The proposed PPL performs well on both semi-supervised and fully-supervised semantic segmentation and the experimental results are promising.
3. The proposed method achieves the state-of-the-art results of semi-supervised semantic segmentation on both Cityscapes and PASCAL datasets.

* Weakness
1. This paper is motivated to explore the interactions between the pixels and negative prototypes for the following two reasons, i.e., contexts and robustness. However, the proposed two techniques, i.e., contrastive-based module and attention-based module, will simultaneously perform the interaction between pixels and positive/negative prototypes. There is no significant clue that the interaction with negative prototypes could benefit from the two above reasons. The multiple positive prototypes, concretely, the temporal positive prototypes, also bring more contextual information and robustness to the noise. For me, current experimental results cannot support the (main) claim well.
2. Although the authors discuss the difference between the proposed PPL with [1,2], the authors also claim that pulling interaction is approximately performed across the dataset, the difference of the attention-based module between this paper and [1,2] is minor. And the contrastive learning between pixels and prototypes has also been explored in [3]. The authors should provide more detailed comparisons with previous similar works, which might limit the novelty of this paper.

[1] Jin et.al. Mining Contextual Information Beyond Image for Semantic Segmentation. ICCV 2021.
[2] Zhou et.al. Regional Semantic Contrast and Aggregation for Weakly Supervised Semantic Segmentation. CVPR 2022.
[3] Zhou et.al. Rethinking Semantic Segmentation: A Prototype View. CVPR 2022.


**Summary Of The Paper:**

This paper presents a Push-and-Pull Learning (PPL) semantic segmentation, which focuses on the interaction between features and prototypes (similar to classifier weights). The proposed PPL consists of a contrast-based pushing module to pull close/push away features according to the contrastive loss between features and CM (classifier memory). The proposed PPL also contains an attention-based pulling module to absorb contextual information from the classifier memory through attention. This paper extends the proposed PPL into a multi-stage cascade structure and incorporates it into a teacher-student framework for semi-supervised segmentation. The experiments on PASCAL VOC and Cityscapes under semi-supervised settings show some good results.

**Summary Of The Review:**

This paper presents a semi-supervised semantic segmentation with the proposed Push-and-Pull Learning. The proposed method achieves promising results under semi-supervised settings. However, the technical novelty of this paper is somewhat limited. Moreover, the main claim of this paper about the negative prototypes has not been well investigated and current experiments cannot demonstrate it and it’s unclear what exactly works in this Push-and-Pull Learning. In all, I’d like to upgrade my score if the authors can better support this claim and clearly state the difference with previous methods using similar methods.

---

> ### Comment · Reviewer_ewoj · 2022-12-06
> **The authors did not provide a response to all reviews.**
>
> The authors did not provide a response to all reviews. After reading comments from other reviewers, I agree that the technical novelty of this submission should be clarified. Further, the proposed pull-and-push learning is not consistent with the claim, and the experiments cannot support the claim well. Several reviewers are also concerned about the insufficient experiments. Consequently, I’m inclined to lower my rating.

---

### Official Review · Reviewer_gCrj · 2022-10-24

**Confidence:** 3
**Correctness:** 4
**Technical Novelty And Significance:** 3
**Empirical Novelty And Significance:** 2
**Recommendation:** 5

**Clarity, Quality, Novelty And Reproducibility:**

- Clarity: borderline
- Quality: borderline
- Novelty: borderline
- Reproducibility: good

**Strength And Weaknesses:**

## Strengths
- The motivation of the paper is clear. The paper contributes a new perspective, i.e., the feature-prototype interaction, and proposes to leverage the historical classifier vectors to enhance the model.
- The proposed methods are simple. It is easy to follow and reproduce the results.
- Clearly better semi-supervised semantic segmentation results are achieved over the baseline and previous methods on PASCAL VOC and Cityscapes.

## Weaknesses
- Figure 5 with only one example is not sufficient to demonstrate the effectiveness quantitatively. More examples should be provided.
- It's not clear whether the methods are still beneficial when transferring to more challenging datasets, e.g., ADE20K and COCO, and stronger baselines, e.g., Swin Transformer.
- More discussion and analysis are needed to explain why the proposed method works. In addition, the limitation of the method is not discussed.
- Why only the negative classifiers are used? How could we also leverage the positive classifiers?
- Why the most of the improvements still hold when the classifier memory bank length T is set to 1?

**Summary Of The Paper:**

The paper studies the interaction between the feature vectors and classifier vectors (i.e., prototype) to improve the performance of semi-supervised semantic segmentation. Specifically, the paper proposes Push-and-Pull Learning (PPL) which consists of a multi-prototype contrastive loss and an attention-based pulling block. The former pushes the current feature vectors to be far from the negative prototypes in the memory bank. The latter adjusts the current feature vector by fusing it with the negative prototypes via a sequential of attention and convolutions. The experiments are conducted using semi-supervised learning settings on two semantic segmentation datasets, PASCAL VOC and Cityscapes.

**Summary Of The Review:**

The paper provides a new perspective on leveraging historical classifiers via contrasting features with historical classifier embeddings. But I still have several concerns listed above. I would rate it as borderline at this stage.

---

### Official Review · Reviewer_xCik · 2022-10-25

**Confidence:** 5
**Correctness:** 3
**Technical Novelty And Significance:** 2
**Empirical Novelty And Significance:** 2
**Recommendation:** 5

**Clarity, Quality, Novelty And Reproducibility:**

The designs are a bit complicated. Without the provided implementations, the reproducibility would require an experienced expert to achieve.

**Strength And Weaknesses:**

**Strength**
* The paper is easy to follow
* The idea of considering the relationship between the feature and prototypes of every category is interesting for semi-supervised learning
* Results show improvement by including the proposed modules for push and pull effect
* Final performance is competitive with existing methods

**Weakness**

Technical novelty and motivation
* One of the main concerns of this paper is the technical novelty, where the contrastive loss has been commonly used for many tasks, e.g., works mentioned in the related work section. It's not clear how the proposed method in Section 3.1 is different from these methods.
* For Section 3.2, the authors have mentioned many times that the proposed method would try to pull the features closer to negative prototypes. However, it appears that in Eq(3), the process is to aggregate features with the prototypes, weighted by the feature-prototype attention scores. This is actually very different from the statements claimed in the paper, and thus some of the motivations in the proposed method is not valid.

Technical clarity and choice
* For the contrastive loss in Eq(1), have the authors considered the infoNCE based formulation to further improve it?
* f_1 and f_2 on Fig 2 are not clearly defined
* The paper mentioned that the memory module CM is updated FILO. However, it's usually designed in a FIFO fashion to keep up-to-date samples. The authors may explain this and provide additional results.

Experimental results
* Experimental results only show limited performance gains over existing methods, e.g., many of the entries are less than 0.5% gain in Table 1.
* For Cityscapes, the authors use CutMix augmentation and OHEM loss. It is not clear whether these boost the performance a lot.
* In the ablation study, Table 3, it's not clear about the effect of using the refined feature in Eq(3). The authors should ablate the push and pull effect in Eq(1), and then also ablate whether the feature is refined in Eq(3).
* In Fig 4, it's not clear to see the advantage of using the 2nd stage of PPL. With more than 1 stage, which proposed component provides more performance gains?
* It would be interesting to know the overhead of training time when using more than one stage PPL.

**Summary Of The Paper:**

The paper proposes a method for semi-supervised semantic segmentation, via a push-and-pull strategy between features and prototypes. Specifically, given a feature, a contrastive loss is used to perform the pull effect for positive prototypes, while the push effect is used for negative prototypes. Moreover, the authors design another module to refine the feature vector by aggregating it with attention-weighted prototype features. This module interact features with prototypes from each category, so the context information can be extracted. To further utilize the proposed scheme into the semi-supervised framework, the same process can be iterated for many times (2 times in the paper) to refine features and pseudo-labels of unlabeled data. Experiments are conducted in the PASCAL VOC and Cityscapes datasets.

**Summary Of The Review:**

Overall, the authors provide an interesting idea of exploring relationships between features and all prototypes of every class. However, the claims are not consistent with the proposed method in Section 3.2. Moreover, it's not clear about the main contribution based on provided experimental results. The authors should carefully address the above-mentioned comments.

---

### Official Review · Reviewer_CAt6 · 2022-10-26

**Confidence:** 4
**Clarity, Quality, Novelty And Reproducibility:** See above for detailed comments on th…
**Correctness:** 3
**Technical Novelty And Significance:** 3
**Empirical Novelty And Significance:** Not applicable
**Recommendation:** 6

**Strength And Weaknesses:**

Strengths:
- The proposed push-and-pull feature refinement strategy seems interesting and novel for the task of semi-supervised semantic segmentation.
- The empirical results show that the method outperforms the previous SOTA methods.

Weaknesses:
- The motivation of this work seems unclear, especially for the task of semantic segmentation. While the proposed idea is interesting, it is rather orthogonal to the segmentation problem. It is unclear why the proposed method, which is essentially a general SSL method, was only applied to semantic segmentation, and what segmentation property has been exploited here.
- The multi-prototype and attention-based feature refinement has been explored in the few-shot semi-supervised semantic segmentation, which is related.
Ling et al,  Semi-Supervised Few-shot Learning via Multi-Factor Clustering, CVPR 2022.
- Some part of the paper is less clear and lacks details:
1) Attention-based pulling: It is unclear what specific type of "contextual information" is captured since the current description is slightly vague.
2) The PPL classifiers used in Equations 5 and 6 (first term) are less clear.
3) The settings for the ablation study are unclear. What is the baseline? How the PPL is trained in the fully-sup setting?
- There are a few concerns on the experiments:
1) The method adds multiple stages of PPL, which essentially leads to deeper networks and deep supervision for multiple stages. What is the model complexity of this design compared to the previous methods and the baseline? It would be interesting to see a comparison of network design with other types of feature refinement, such as the non-local neural network, and/or with deep supervision.
2) It is unclear how the conclusion (Sec 4.3 & 4.4) of " the task-specific benefit is the main reason" was reached based on the results as both modules make important contributions to the final improvement.
3) The difference between one-stage and two-stage PPL seems rather small according to fluctuation in the performance curves. How significant is the improvement from the two-stage PPL?

**Summary Of The Paper:**

The paper presents a semi-supervised learning method for semantic segmentation, which learns a multi-prototype representation for each semantic class and a feature refinement step based on those prototypes. To achieve this, it develops a multi-prototype contrastive loss function and an attention-based prototype-to-feature fusion module. The proposed feature learning strategy is then integrated into a teacher-student framework, in which the student network is augmented with the feature refinement modules and trained with a hybrid loss consisting of cross-entropy losses on labeled data and pseudo-label of unlabeled data, and the contrastive loss on unlabeled data. The authors evaluate the method on Pascal VOC and Cityscapes benchmark with comparisons to the prior art.

**Summary Of The Review:**

The proposed feature refinement strategy seems interesting and effective for the SS task. However, there are several concerns on the lack of clarity in motivation and presentation, and some missing ablation studies.

---

> ### Comment · Reviewer_CAt6 · 2022-11-28
> **No rebuttal provided**
>
> The authors did not provide feedback. Due to the lack of clarity and the missing ablation studies, the overall results seem less convincing and I would lower my rating accordingly.

---

### Decision · Program_Chairs · 2023-01-20

**Decision:**

Reject

**Justification For Why Not Higher Score:**

There were several issues needed to be addressed by the authors. However, no rebuttal was provided.

**Justification For Why Not Lower Score:**

N/A

**Metareview: Summary, Strengths And Weaknesses:**

The paper proposes a new perspective for semi-supervised semantic segmentation by modeling interactions between features and prototypes. A prototype is the weight vector (in the classification head) for each class. Typically, the approaches in this domain pull features and positive prototypes towards each other and push away the negative prototypes. Contrary to existing approaches, the paper proposes to pull close negative prototypes as it is beneficial for the final performance. An attention mechanism is used to pull close negative prototypes.

Strengths:
- The feature refinement strategy is interesting and provides a new perspective.
- The results are competitive with the results of the existing SOTA methods.
- The proposed method is simple, and the paper is easy to follow.

Weaknesses:

There are various issues including:
- The reviewers are concerned that the claims in the paper are not well supported – reviewers ewoj and xCik
- There are novelty concerns compared to previous work – reviewers ewoj and xCik
- There is lack of clarity about the contribution of different components – reviewers CAt6 and xCik

The authors did not provide a rebuttal to respond to the questions raised by the reviewers. So, there are several issues that need to be clarified. The majority of the reviewers are leaning towards rejection, the AC agrees with the reviewers’ concerns and recommends rejection.